

# Interdecadal rainfall cycles in spatially coherent global regions and their interaction with climate modes

Tobias F. Selkirk[1], Andrew W. Western[1] and J. Angus Webb[1]

[1]Department of Infrastructure Engineering, University of Melbourne, Parkville, 3052, Australia

*Correspondence to*: Tobias F. Selkirk (selkirkt@unimelb.edu.au)

**Abstract.**

Interdecadal cycles in rainfall influence long-term hydrological variability, affecting water resource management, agriculture, and flood or drought preparedness across the globe. Previous studies have found evidence of cycles over limited
regions but the global distribution and interaction with major climate modes remain unclear. Using the global GPCC v2022 2.5° gridded dataset (1891-2020), we applied a Gaussian mixture model to detect significant clustering of cycles in rainfall, derived from wavelet analysis of individual grid points. Three Global Rainfall Cycles (GRCs) emerged at 12.9-, 19.9-, and 28.2-years, were widespread, and aligned in length and phase to previous research. Two longer cycles (35.9- and 45.9-years) were also significant but interpreted cautiously due to their period relative to the dataset's length. The 12.9- and 19.9-year
GRCs showed strong phase coherence and spatial overlap with the El Niño-Southern Oscillation and Interdecadal Pacific Oscillation climate modes, but not with the Indian Ocean Dipole or North Atlantic Oscillation. Notably, GRCs explained more rainfall variance than expected from the effect of these climate modes alone, suggesting another driver may influence rainfall directly and via climate interactions. These findings are of significance to global water management and rainfall modelling, offering the potential to enhance flood and drought forecasting in strongly affected regions.

## 1 Introduction

Understanding long-term hydrological variability associated with interdecadal cycles in global rainfall underpins effective water resource management, agricultural planning, and resilience to floods and droughts. These cycles, in the range of 10–50 years, are often subtle and challenging to detect due to their low amplitude relative to seasonal and interannual variability, as well as the limitations of historical rainfall datasets (Sun et al., 2018).

Interdecadal periodicity in rainfall has been studied extensively, though mostly limited to specific regional areas (Chowdhury and Beecham, 2012; Kane, 2009; Williams et al., 2021; Selkirk et al. 2025), and within in the major climate modes (An and Wang, 2000; Sun and Yu, 2009; Yasuda, 2018). The global distribution of climate mode influence on rainfall is also well established (Baines, 2011; Becker et al., 2013; Cayan et al., 1998). However, we are aware of no studies that have attempted to identify dominant periodicities in globally complete rainfall datasets and their spatial relationship to

the climate modes. This is likely due to the complexity of separating interdecadal signals from noise in diverse climate





regimes, scarcity of long-term high-quality global rainfall data and the challenges of processing large datasets using existing signal decomposition techniques. Consequently, the global extent of these cycles, their spatial distribution, and their potential drivers remain poorly understood.

A recent study in eastern Australia identified three dominant interdecadal cycles (~13-, ~20-, and ~28-years) present at a majority of sites, demonstrating their regional importance (Selkirk et al., 2025). The methodology used varied from the traditional approach of defining significance by the individual power of each cycle over random noise at a single site (Grinsted et al., 2004; Murumkar and Arya, 2014; Torrence and Compo, 1998). Instead, significance was derived from clustering the number of sites at which common cycles occurred across a large dataset. This allowed for the identification of cycles with a subtle influence, but present at nearly all sites.

Many regional studies have identified interdecadal periodicities in rainfall often clustered around ~13 and ~20 years. These two cycles have often been attributed to the ~11-year Sunspot cycle, and the 18.6-year lunar nodal cycle (LNC) correlated to rainfall in India (Mitra and Dutta, 1992), Australia (Currie and Vines, 1996), the Americas (Currie, 1983, 1984), Africa (Currie, 1993), Mongolia (Davi et al., 2006) and Russia (Currie, 1995). Synchronisation of rainfall to the lunisolar drivers tended to fall out of alignment over time, requiring a phase inversion roughly every 100 years. This

generated scepticism and restricted broad academic acceptance of the presence of these cycles in rainfall.  Selkirk et al. (2025) discovered that, in Australia at least, the need for phase inversion was a result of focusing on the incorrect cycle length. Alignment to slightly longer  of 13.1- and 19.9-years were consistent across the whole 130-year time series.

        Cycles around this length continue to be observed in regional rainfall, though are rarely ascribed to a stable driver. They are often explained through the established climate modes known to drive rainfall variability on interannual to decadal

scales. These include the El Niño-Southern Oscillation (ENSO), Interdecadal Pacific Oscillation (IPO), the Indian Ocean Dipole (IOD) and North Atlantic Oscillation (NAO). Williams et al., (2021) identified a significant 13 to 15-year cycle in Sierra Nevada, accounting for 21% of rainfall variability from 1902 to 2000. Tree ring reconstruction dating back over 600 years also indicated 12.8- and 21.3-year cycles with marked amplitude variance over time. However the authors did not believe these cycles were the tied to ENSO variability. An 11 to 13-year cycle was identified in the Ohio River Basin,

strongly correlated to ENSO (Amonkar et al., 2023) and a ~20-year oscillation in South African rainfall has also been attributed to the climate mode (Kane, 2009).

        Most global rainfall analyses interpret interannual variability through the lens of the major climate modes (Baines, 2011). Sea Surface Temperatures (SST) often form the quantitative foundation of these modes, however they involve the complex interactions of multiple dynamic atmospheric and oceanic systems. This results in global influences tied together

though the push and pull of atmospheric teleconnections. For example, during the La Niña phase of ENSO, cooler SST in the Niño 3.4 region of the Pacific Ocean intensify the Walker Circulation winds, leading to stronger convection over the warmer western waters and driving increased rainfall over eastern Australia. On the other side of the Pacific, the sinking air inhibits cloud formation and precipitation leading to drier conditions along the west coast of the Americas (Cai et al., 2011).



The interdecadal shifts in ENSO are more clearly represented in the IPO, which is generally understood to be a long term modulator of its influence (Power et al., 1999). The index is derived from empirical orthogonal functions (EOFs) of low-pass-filtered (11-year) SST anomalies in the pacific (Parker et al., 2007). Variability within the 10–30-year band is often assumed to be quasi-periodic and unstable, varying in amplitude and frequency over time (Lorenzo et al., 2023). Some studies, however, suggest greater stability, identifying consistent ~18.6-year cycles in the Pacific Decadal Oscillation (PDO) and ENSO, potentially linked to the LNC (Yasuda, 2009, 2018). Understanding the global influence of these interdecadal cycles is essential for improving climate models and predicting long-term hydrological trends, particularly in regions vulnerable to water scarcity and extreme events.

This study sought to identify and quantify interdecadal rainfall cycles globally, using the Gaussian Clustering of Wavelet Amplitude Power Spectrum (GC-WAPS; Selkirk et al. 2025) method applied to the GPCC v2022 dataset (1891–2020). We aimed to identify Global Rainfall Cycles (GRCs) and assess their spatial distribution, phase alignment, and correlation with major climate modes. Our results reveal the global presence of the three cycles (~13-, ~20- and ~28-years), with coherent phase alignment in regions like Australasia and the Americas driven partially by the El Niño Southern Oscillation. However, the climate modes cannot fully account for the scale of influence observed.



## 2 Study Area and Data

The global dataset used was from the Global Precipitation Climatology Centre (GPCC), established by the World Meteorological Organization in 1989 and operated by Deutscher Wetterdienst (National Meteorological Service of Germany). The GPCC Full Data product is based on in situ rain gauge data from over 85,000 stations and is the most commonly used global precipitation dataset for climate variability research as it incorporates many sources from international regional projects (Sun et al., 2018). The GPCC Full Data Monthly Product Version 2022 was used at a 2.5°

gridded resolution comprising 3,638 land surface data points covering all years from 1891 to 2020. Annual rainfall at each grid point was calculated by summing the monthly data for each calendar year. The effect of spatial interpolation in eastern Australia was tested against the infilled daily station data from 1889 to 2022 via the SILO database, hosted by the Queensland Department of Environment and Science (Jeffrey et al., 2001), also summed by calendar year.

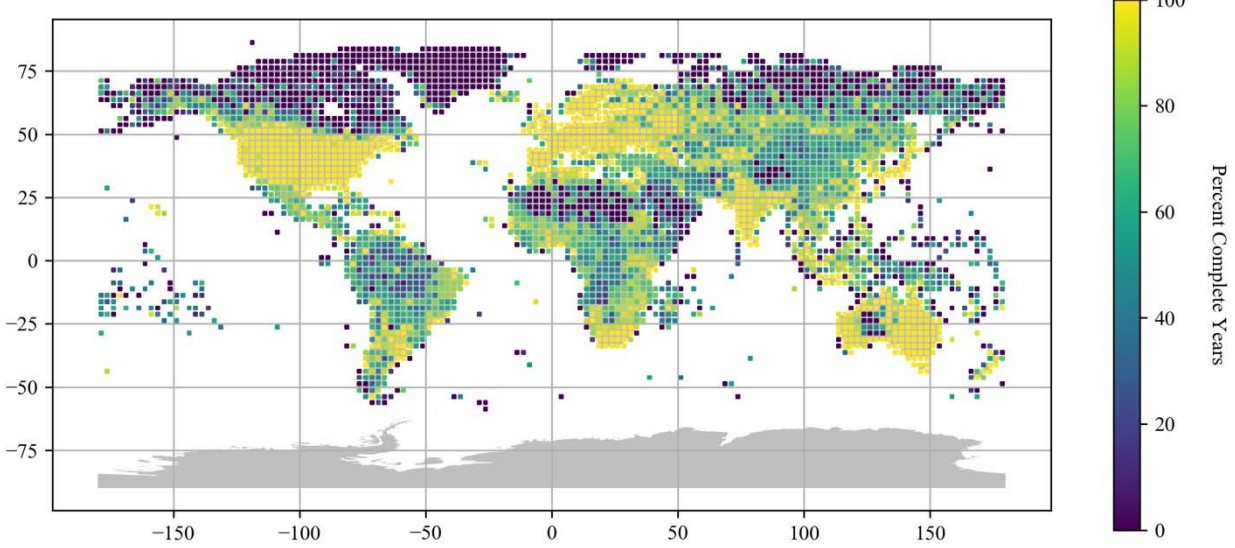

**Figure 1.** GPCC v2022 dataset precipitation grid points coloured by the percentage of years in which all monthly time series measurements are informed by at least one station gauge.

GPCC v2022 makes substantial use of infilling by the insertion of climatological normals where an entire 5° grid is without station data. The dataset also includes monthly values on the number of gauges informing each grid point. To

determine the percentage of years with sufficient gauge coverage for each grid point, we identified the percentage of complete years where all months were informed by one or more gauges (Fig. 1).

Several indices were chosen to characterise various climate cycles. The Niño 3.4 index was chosen over composite ENSO indices (e.g., Southern Oscillation Index) due to its simpler, cleaner signal and stronger correlation with global rainfall. It consists of area-averaged sea surface temperature (SST) anomalies over the Niño 3.4 region (5°N–5°S, 170–



120°W) from 1870 to 2025. It was sourced from the NOAA Physical Sciences Laboratory (2017) and is calculated using the ERSST v5 dataset (Huang et al., 2017). Monthly values were averaged by calendar year. The Interdecadal Pacific Oscillation (IPO) index, representing annual variation in decadal-scale Pacific SST variability (1871 to 2016) was sourced from the New Zealand Ministry for the Environment (2017) .

The North Atlantic Oscillation (NAO) index represents the atmospheric pressure variability between the Subtropical
(Azores) High and the Subpolar Low. Monthly values from 1950 to 2025 were sourced from the NOAA National Centers for Environmental Information (2025) and averaged by calendar year. The Indian Ocean Dipole (IOD) index is a measure of the anomalous SST gradient between the western (50°E–70°E, 10°S–10°N) and south-eastern (90°E–110°E, 10°S–0°) Indian Ocean. Monthly values from January 1871 to January 2025 were sourced from the NOAA Physical Sciences Laboratory (2017), calculated using the ERSST v5 dataset (Huang et al., 2017). Values were averaged by calendar year and the time
series was detrended using linear regression.

## 3 Methods

### 3.1 Gaussian Clustering of Wavelet Amplitude Power Spectrum (GC-WAPS)

To identify interdecadal cycles in global rainfall, we employed the Gaussian Clustering of Wavelet Amplitude Power Spectrum (GC-WAPS) method, a novel approach for detecting significant periodic signals in large sets of time series
data (Selkirk et al., 2025). GC-WAPS applies a continuous wavelet transform to decompose rainfall time series into frequency components and generates a global mean power spectrum (GMPS) by averaging the absolute wavelet coefficients across the time series. Peaks in the GMPS at each site are automatically selected and collated. A Gaussian mixture model (GMM) is used to identify clusters of periods with a density higher than expected from generated red noise. Within each cluster the values are subtracted from the cluster mean, and the significance calculated as indication of spread relative to red
noise by $t$ test.

The previous application of GC-WAPS was able to use high-quality rain gauge data to detect subtle interdecadal cycles in eastern Australia (Selkirk et al., 2025). In the current study, the GPCC v2022 dataset includes substantial climatological infilling (Fig. 1). Periods of missing data manifest as flat segments in the time series and can hinder the detection of correct periodicity. To ensure reliable cycle identification, we filtered the dataset to include only grid points with
at least 90% of years from 1891 to 2020 (130 years) having all months informed by at least one gauge. This reduced the number of usable grid points by 75%, from 3,638 to 909, with coverage concentrated in Australia, Europe, India, and North America (Fig. 1).

To ensure trends from the changing global climate did not introduce harmonic distortion into the wavelet spectrum (Stéphane, 2009), we applied the augmented Dickey-Fuller test using the *statsmodels* package (Seabold and Perktold, 2010)
to detect non-stationarity in the rainfall data at each grid point. Sites with significant trends ($p < 0.05$) were detrended using a linear least-squares fit with the *SciPy detrend* function (Virtanen et al., 2020) before wavelet analysis. The data were




normalised by variance ($1/\sigma^2$) to ensure the wavelet transforms at each site were directly comparable (Torrence and Compo, 1998).

GC-WAPS applies a continuous wavelet transform using the *Morlet* wavelet function, with a bandwidth of 6 and a centre frequency of 1. The *PyWavelets* package (Lee et al., 2019) was used for signal decomposition, and the *scaleogram* package facilitated visualization of the wavelet transform. Cycle periods were analyzed from 1 to 80 years in 0.1-year increments. The wavelet function extends past the analysed time-series as it approaches the boundary and this extension is padded with zeros in the data. This makes the magnitude of the spectrum within this region less reliable, but periodicity can still be identified. For a *Morlet* function, the COI shortening at each end of the series is proportional to the period by a factor of √2 (Torrence and Compo, 1998) giving a ~50-year practical upper limit. An 80-year cycle is well beyond the Cone of Influence (COI) for a 130-year time series however the extended range was used as a scan for longer periods in global rainfall that may warrant further investigation.

Prominent cycles were extracted using the *SciPy Signal* module *Find Peaks* function (Virtanen et al., 2020), clustered with GMM using the *scikit-learn* package (Pedregosa et al., 2011), and tested for significance against red noise using a *t* test (Selkirk et al., 2025).

The GPCC v2022 dataset is derived from underlying gauged data at individual sites, spatially interpolated to a 0.25° resolution using a modified SPHEREMAP empirical interpolation method, and then aggregated to the 2.5° resolution used in this study via spatial averaging (Schneider et al., 2022). To validate the GPCC v2022 data and assess the impact of aggregating station data on cycle identification, we filtered the gridded dataset to the eastern Australia region and compared the results with those from the SILO gauged dataset (Selkirk et al., 2025). Similarly, the GPCC data were also tested across three longitudinal sectors: The Americas (-150° to -30°), Europe and Africa (-30° to 60°), Australasia (60° to 180°). The entire GC-WAPS method was repeated on data filtered to these three sectors to test whether the patterns were consistent between these regions.

A sliding-window Pearson correlation was computed between the observed signal and a reference cycle (sine wave of matching period) across a range of lags. Within each predefined window corresponding to a single period of the cycle, the year yielding the highest correlation was recorded as the best-fit offset. This approach identifies the lag that maximizes linear similarity between the observed and reference signals, providing a localized phase estimate. Phase offsets from all sites were aggregated into a histogram for each cycle length. Each histogram typically exhibited two peaks: one corresponding to signals in phase, and the other approximately 180° out of phase (i.e., T/2 offset from the reference period T).The results of the optimised phase for the Australasian region were compared to those of Selkirk et al. (2025) for consistency. All analyses were undertaken in Python.

## 3.2 Individual Site Analysis

Individual site analyses were conducted using the full GPCC v2022 dataset (3,638 grid points) to quantify the presence and strength of the Global Rainfall Cycles (GRCs) at each site, overcoming the limitations of the filtered dataset





used in GC-WAPS (Sect. 3.1). The filtered dataset (909 grid points) was too restrictive for assessing global influence, and the GC-WAPS method's sensitivity to data quality posed challenges with low-resolution data. In wavelet analysis, a time series missing or infilled data can reduce cycle power, leading to false negatives, while noise can broaden and merge peaks, causing false positives in cycle length extraction. To address this, we developed two complementary methods to detect and characterize the GRCs across all sites.

First, we tested each site's time series for non-stationarity using the augmented Dickey-Fuller test from the *statsmodels* package (Seabold and Perktold, 2010), detrending sites with significant trends ($p < 0.05$) using a linear least-squares fit via the *SciPy detrend* function (Virtanen et al., 2020). We then extracted the significant GRC periods (12.9-, 19.9-, 28.2-, 35.9-, 45.4-year; Sect. 4.1.1) from the wavelet transform at each site. Second, to assess cycle consistency, we also generated a sine wave for each GRC period, phase aligned to the Australasian region.

The Pearson correlation coefficient (R) and associated p-value were calculated using the *pearsonr* function from the *SciPy stats* package (Virtanen et al., 2020) to evaluate cyclicity in three ways: (1) correlation between the extracted wavelet and rainfall anomalies, providing a direct measure of each cycle's contribution to rainfall while accounting for frequency and amplitude modulation; (2) correlation between the fixed sine wave and rainfall anomalies, indicating the stability of the cycle's influence on rainfall; and (3) correlation between the fixed sine wave and the extracted wavelet, assessing phase

alignment and cycle stability. The null hypothesis was that the distributions are uncorrelated and normally distributed. The coefficient of determination ($R^2$) was calculated to estimate the variance in rainfall explained by each cycle. Additionally, the signal-to-noise ratio (SNR) was computed as the peak amplitude of the extracted wavelet divided by the site's standard deviation ($\sigma$), providing a measure of cycle amplitude relative to background variability. This approach enabled detection of the GRCs even at sites with low temporal coverage.


### 3.3 Global Distribution

    To visualize and quantify the global distribution and spatial clustering of the GRCs, we analyzed the full GPCC v2022 dataset, experimenting with various correlation thresholds to optimize the representation of cycle influence. We filtered sites based on the Pearson correlation coefficients (R) calculated in Sect. 3.2, retaining only those exceeding

specified thresholds: 0.1 for rainfall correlations, representing at least 1% of rainfall variance ($R^2 \geq 0.01$), and 0.25 for wavelet correlations, selected through experimentation to best visualize spatial coherence and eliminate random phase mixtures. This process largely removed regions with random mixtures of positive and negative correlations in adjacent sites, highlighting areas of coherent phase alignment and high correlation. For each GRC period, we calculated the number of sites above the threshold, the mean variance explained by the cycle ($R^2$) from both the fixed sine wave ($\bar{x}_{var\,(fixed\,)}$) and extracted

wavelet ($\bar{x}_{var\,(wavelet\,)}$), as well as the mean signal-to-noise ratio ($\bar{x}_{SNR}$) across these sites, providing overall metrics of the cycles' global influence.





**3.4 Climate Modes**

To visualize the global impact of climate modes, we applied a method similar to that used for the GRCs in Sect. 3.2.

The Pearson correlation coefficient (R) and associated p-value were calculated between the annual rainfall anomaly at each site and the annual time series of each climate mode index, using the *pearsonr* function from the *SciPy* stats package (Virtanen et al., 2020). Sites were filtered using the same correlation thresholds as in Sect. 3.3. The Global Mean Power Spectrum (GMPS) was generated for each climate mode index by averaging the absolute wavelet coefficients across the time series, following the wavelet analysis procedure in Sect. 3.1. Prominent cyclic components were identified using the *SciPy*

*Signal* module's *find_peaks* function (Virtanen et al., 2020). For climate mode cycles with periods close to the GRCs (12.9-, 19.9 and 28.2-years), the wavelet was extracted at those periods and compared to a fixed sine wave phase aligned to the Australasian region.





# 4 Results

**4.1.1GMM Clusters of the Global Rainfall Cycles (GRCs)**

Three significant cycles (p < 0.05) were found in the global data, with clusters centred around 12.9-, 19.9- and 28.2-year (Fig. 2). These cycles are similar to those previously observed in eastern Australia using the same method (13.1-, 20.4- and 29.1-year; Selkirk et al., 2025). However, the density of sites within each Gaussian distribution was not as high, leading to less prominent clustering (Fig. S1b in the Supplement). This is consistent with our expectations, as the ~13- and ~20-year

cycles were nearly ubiquitous across all Australian sites, a pattern not assumed to hold globally.

The results of the red noise replication of GC-WAPS method (grey outline, Fig. 2b) exhibit a gradual decline in density with increasing cycle length, consistent with the expected spectral characteristics of a first-order autoregressive AR(1) process. This indicates the clustering observed in the GPCC data (filled grey, Fig. 2b) is not an artefact of the analytical method.

Two other cycles of longer periodicity emerge in the global analysis with means centred at 35.9- and 45.4-year. Though these two cycles are significant (p<0.05) they appear at a relatively small number of sites. They are also approaching the upper limit of periodicity (~50-year) that can be justified by wavelet analysis from a 130-yeartime series. Although their presence is noted, it is also treated with some caution.

Breaking the results down into three broad sectors (Fig. 3) allowed us to test whether the patterns were consistent

across longitudinal geographic sectors. The ~13-year cycle remains significant (p<0.05) across all three regions a with slight variation in its mean (from 11.5 to 13.2-years). Such variability is expected since wavelet analysis does not provide a high degree of precision with regard to cycles, especially in noisy data.

Clustering around the ~20- and ~28-year periods is evident in the underlying histograms (grey filled) for all regions when compared to random red noise, but is sometimes slightly below the significance threshold (p = 0.1, 0.14). This suggests

that while the cycles may be less dense in these regions, they are nonetheless present. Regional comparison reveals the highest cycle density in Australasia, with consistent evidence of the same cycles across all regions and no emergence of alternative periods.





**Figure 2.** Results for the cycle selection from GC-WAPS at 909 sites. (**a**) location of all data points with ≥90% of time series informed by at least one gauge record within each 2.5° grid cell. (**b**) GMM clustering of cycles significant against a red noise background (*t* test, $p < 0.05$). The filled light grey histogram represents all periods extracted from the filtered GPCC v2022 annual rainfall data by wavelet analysis. The grey outline is derived from the same process repeated on red noise, showing an even decay in power with no apparent clustering. Significant clustering occurred around cycles highlighted in blue (12.9-year), red (19.9-year) and green (28.2-year). Two additional cycles of 35.9- and 45.4-years were also found to be significant ($p < 0.05$).





**Figure 3.** Separation of rainfall data into three longitudinal segments. (**a**) The Americas (-150° to -30°); the ~13-year cycle is significant (p < 0.05), with clustering of cycles (pale grey histogram) at ~20- and ~28-year marginally non-significant (p > 0.05). (**b**) Europe and Africa (-30° to 60°); the ~13-year cycle is significant (p < 0.05), with clustering at ~20- and ~28-year also marginally non-significant (p > 0.05). (**c**) Australasia (60° to 180°); all three cycles (~13-, ~20-, and ~28-year) are present and highly significant (p < 0.05).





## 4.2 Individual Site Analysis

We expanded the analysis to all sites using the precise mean values for GRCs (12.9-, 19.9-, 28.2-year) phase
aligned to the Australasian region. An incomplete time series may obscure significant peaks in the GMPS (Fig. 4), yet their
presence can still be detected using more direct methods. By way of example, wavelet analysis at one chosen site in South
America with only 26.2% of gauge-informed monthly values reveals a broad concentration of power between 15 to 35-year
periods, marked by a large red zone, with the GMPS (Fig. 4b, d) showing its largest peak at 24.7-years. The automated peak
finder, indicated by vertical red lines, identifies only this single peak between 10 and 40 years. Yet, extracting the 12.9-,
19.9-, and 28.2-year cycles (orange) reveals they are closely in phase with those in the Australasian region (green). The 19.9-
year cycle exhibits a strong amplitude, with a high signal-to-noise ratio (SNR) of 1.11 and an extracted wavelet accounting
for 16% of annual rainfall variance ($R^2$). This demonstrates that, while incomplete time series are unsuitable for GC-WAPS
analysis, they can effectively confirm the presence and phase of these cycles through targeted extraction, enabling a clearer
global distribution to be found that is not limited to regions of high-quality data.

This can be contrasted with a second example using a complete time series (100% gauge-informed monthly values)
from the Sierra Nevada in North America (Fig. 5). Here, three cycles at 14.3-, 21.1-, and 29.6-year are identified in the
GMPS (Fig. 5d) but are 180° out of phase with the 12.9-, 19.9-, and 28.2-year GRCs (Fig. 5e–g). The slight differences
between these GMPS-derived periods and the GRC periods are attributable to the relatively broad peaks in the power
spectrum.

Individual analyses was also performed on the 35.9- and 45.4-year cycles identified in the GC-WAPS results (Sect.
4.1.1). However, these cycles exhibit notably lower signal-to-noise ratios (< 0.5) compared to the first three, occur less
frequently across the dataset, and approach the upper limit of what can be reliably resolved using wavelet analysis given the
length of the time series. As such, their interpretation carries greater uncertainty.






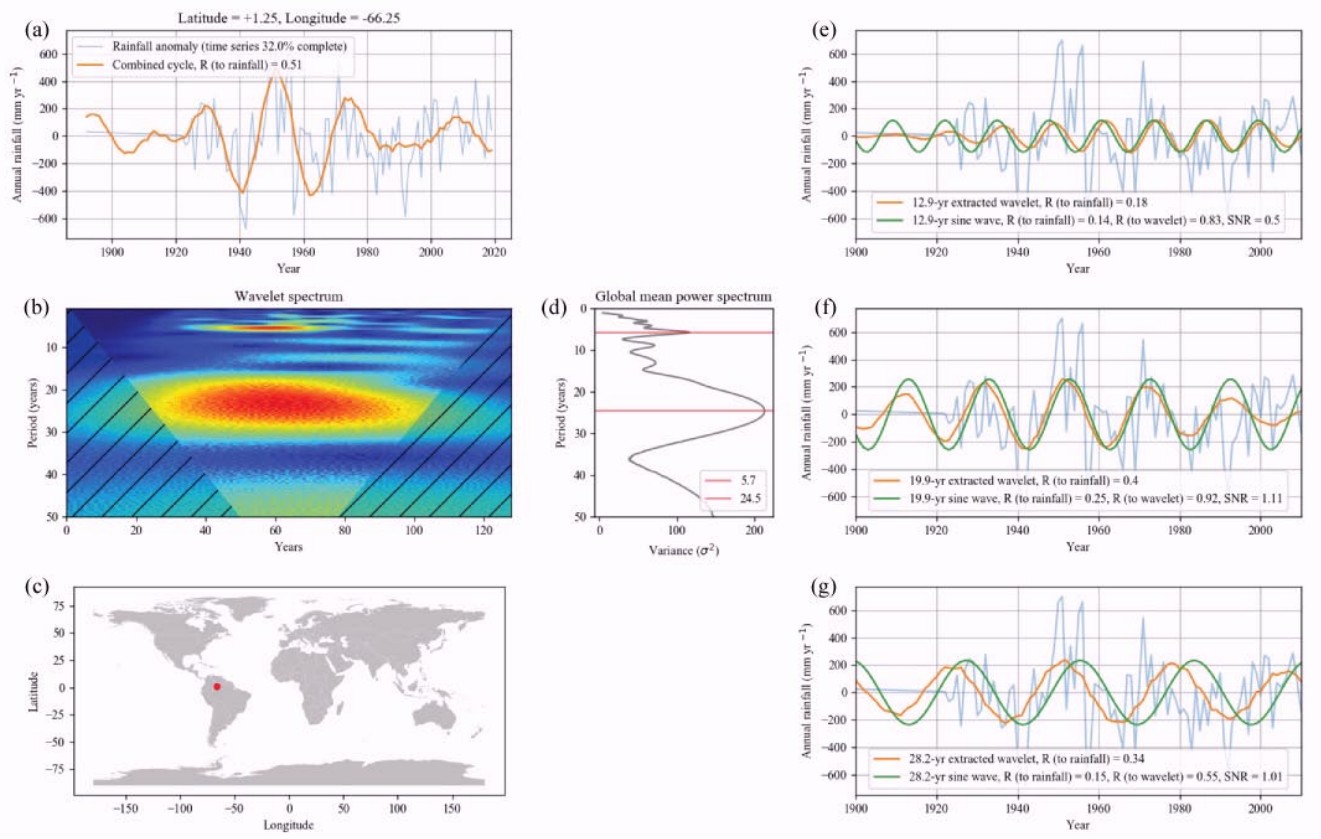

**Figure 4.** Single site analysis in South America near the border of Venezuela and Brazil. (**a**) Time series of annual rainfall (light blue), overlaid with the combined waveform of the 12.9-, 19.9- and 28.2-year cycles. (**b**) The wavelet analysis spectrum. (**c**) Site location on a global map. (**d**) The GMPS of the wavelet analysis with horizontal red lines showing the peaks automatically selected. (**e-g**) individual analysis shown for three main cycles picked up in the global GMM showing annual rainfall anomaly (light blue), fixed cycle extracted from wavelet analysis (orange), and a generated sine wave for each period (12.9-, 19.9-, 28.2-year) phase aligned to the Australasian region. Although the 12.9- and 19.9-year cycles were not detected by the automated GMPS peak selection, the extracted wavelets exhibit near-perfect phase alignment with the GRCs.






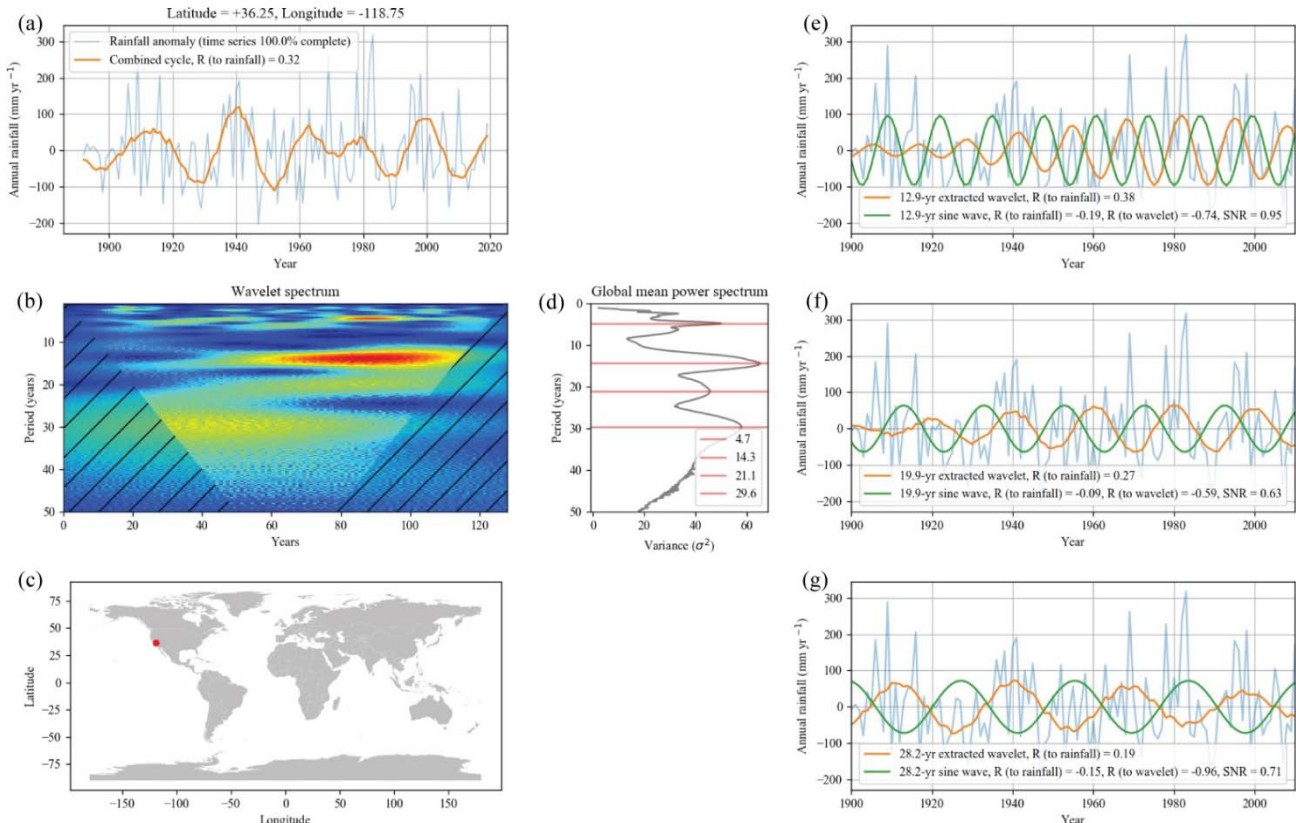

**Figure 5.** Single site analysis in Sierra Nevada, America. (**a**) Time series of annual rainfall (light blue), overlayed with the combined waveform of the 12.9-, 19.9- and 28.2-year cycles. (**b**) The wavelet analysis spectrum. (**c**) Site location on a global map. (d) The GMPS of the wavelet analysis with horizontal red lines showing the peaks automatically selected. (**e-g**) Individual analysis shown for three main cycles picked up in the global GMM showing annual rainfall anomaly (light blue), fixed cycle extracted from wavelet analysis (orange) and a generated sine wave for the chosen period phase locked to the Australasian region. The alignment of the 12.9-, 19.9- and 28.2-year cycles are all approximately 180° out of phase with GRCs.






### 4.3 Spatial Distribution of the Global Rainfall Cycles (GRCs)

This individual site analysis was conducted for all 3,638 sites in the GPCC v2022 dataset, with results aggregated to map the global spatial distribution (Fig. 6). The influence of the GRCs on rainfall at each site exhibits clear spatial coherence, this influence is subtle, yet widespread. The left panels of Fig. 6 (a-c) show the correlation of the GRC's directly to rainfall. However, since the signal of the cycles is often weak (either inherently or due to noisy and incomplete data) a more generous distribution is shown where the extracted wavelet for each cycle is correlated to the fixed sine wave (Fig. 6d-

f). This is more indicative of phase alignment than rainfall contribution, broadly speaking it often shows a wider phase coherence in the region surrounding the significant clusters.

For instance, the 12.9-year GRC shows its spread from Australia through Indonesia into northern Asia, all in phase, while disconnected clusters in northern Africa, Europe, and the Middle East (Fig. 6a). As the threshold is relaxed these disparate areas merge into a near-continuous region in the opposite phase (Fig. 6d). In the conterminous United States, we

can see more detail as the east and west coasts align to opposite phases. The average annual rainfall variance explained at significant 12.9-year GRC sites is 6.19% using the fixed sine wave, rising to 11.16% with the extracted wavelet. An average signal-to-noise ratio of 0.89 hints at a much larger influence, indicating that the average amplitude of the cycle is close to one standard deviation of annual rainfall.

The 19.9-year GRC (Fig. 6b, e) exhibits similar regions of influence. Eastern Australia is in phase, with its

influence extending into Asia and aligning with the east coast of the USA. The west coast of the USA, along with Chile and Brazil, appears in the opposite phase. The average annual rainfall variance explained at significant 19.9-year GRC sites is 5.78% using the fixed sine wave, rising to 10.99% with the extracted wavelet. Again the SNR of 0.91 approaches one standard deviation, indicating that while the interannual fluctuations dominate the total variance, the underlying cycle still exerts a substantial influence.

The 28.2-year GRC shows its greatest influence over Europe, central South America and in the opposite phase over Greenland (Fig. 6c, f). The average annual rainfall variance explained at significant sites is 7.62% using the fixed sine wave, rising to 9.53% with the extracted wavelet. The influence on the Australasian region is notably weaker than the 12.9- and 19.9-year cycles and the in-phase signal clustered closer to the eastern coastline.

Global distributions of the 35.9- and 45.4-year cycles are included in the supplement (Fig. S2). As noted previously

these are treated with some caution. The 35.9-year cycle shows only small regions of significant clustering widely dispersed across the globe. The 45.4-year cycle shows particularly strong influence in the Arctic region of Russia, and is significant at 18% of sites accounting for 11-12% of rainfall variance.





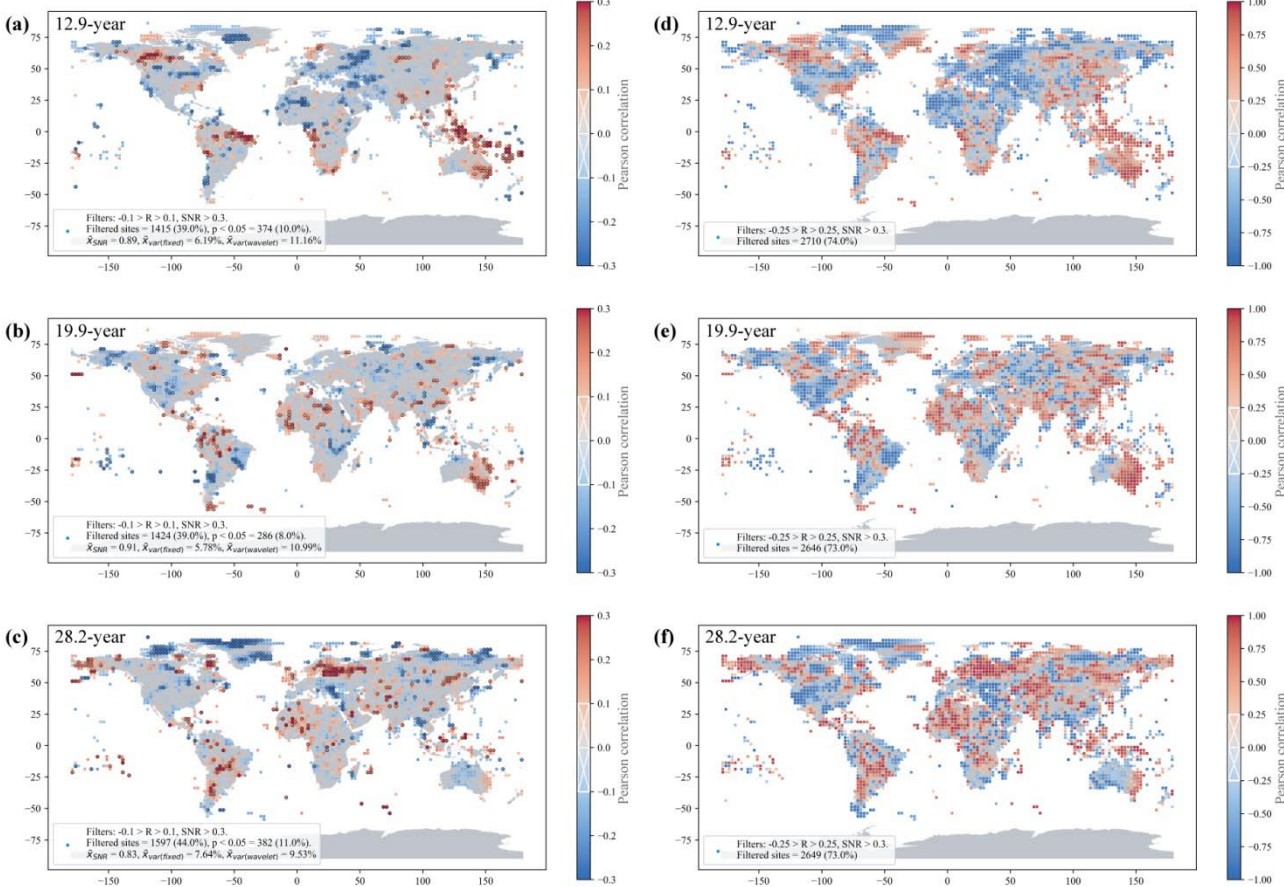

**Figure 6.** Global distribution of the 12.9-, 19.9- and 28.2-year GRCs modelled as sine waves with phase offsets matching the Australasian region. (**a-c**) The Pearson correlation coefficient (R) of the sine wave to rainfall anomaly at each site with significant sites ($p < 0.05$) circled in black, and display threshold of 1% of annual rainfall variance calculated by coefficient of determination ($R^2$). (**d-f**) The Pearson correlation coefficient (R) of the GRC sine wave to the corresponding extracted wavelet with a threshold of 0.25. Both methods show clustering in the same regions for each cycle. Comparison to the wavelet allows a broader picture of the cycle influence by controlling the masking effect of high frequency noise.





**4.4 Relationship to Climate Modes**

To explore potential drivers of the Global Rainfall Cycles (GRCs), we analysed their correlation with major climate modes. Figure 7 displays the spatial distribution and cyclic components while Fig. 8 directly compares the extracted wavelets to evaluate phase alignment.

The IPO shows the strongest spectral alignment with the GRCs, as GMPS peaks at 12.8- and 19.5-year (Fig. 7d), closely match the 12.9- and 19.9-year GRCs. Its influence spans Australasia, North America, and South America (Fig. 7c),

mirroring the GRC's distribution (Fig. 6a, d). Time series analysis confirms this, with the 12.9- and 19.9-year GRCs showing high correlations (R = 0.93) with IPO-derived wavelet cycles, although 180° out of phase (Fig. 8a, b). This is to be expected as the IPO index is focussed on the interdecadal component of SST. The 19.9-year component of the Niño 3.4 index can be seen in the GMPS (Fig. 7e) but the extracted wavelet (Fig 8.b) shows how weak the signal is, contributing only 4% of temperature variance.

The 3 to 7-year quasi-periodicity in the Niño 3.4 index is well established and can be seen in the GMPS (Fig. 7e). However, these results show a secondary peak of higher magnitude at 12.8-year, aligning closely with the 12.9-year GRC. Its influence in Australasia, South America, and Indonesia (Fig. 7a) overlaps with the 12.9-year GRC's distribution, and time series comparison reveals a strong correlation (R = 0.96, inverted, Fig. 8a), indicating that La Niña phases may amplify this cycle in these regions.

We also find similar peaks to the 28.2-year GRC in Niño 3.4 (27.7-year) and the IPO (27.9-year). The correlation is excellent (>0.98), although it requires an ~8-year shift of the 28.2-year GRC, suggesting possible methodological artefacts in the phase calculation (see Discussion). The global distribution of the 28.2-year GRC (Fig. 6c, f) shows several regions of overlap with IPO including the east coast of Australia, Indonesia, Africa and the American West Coast.

Although there are cycles in the NAO and IOD, these bear little relationship to the GRCs. The IOD shows an 11.5-

year and the NAO a 14.1-year peak (Fig. 7f) but neither of these align well to the 12.9-year GRC in phase or period when directly compared (Fig. S3 in the Supplement.). The cluster of IOD influence along the Great Australia Bight (south coast) is not reflected in any of the GRC distributions.

There is a consistent ~45-year cycle across the IPO, IOD and NAO climate modes (Fig.7f-h), similar to the 45.4-year cycle identified by GC-WAPS (Fig. 2). The length of this cycle relative to the time series (~130 years) and its position

near the practical COI limit (~50 years; Sect. 3.1) warrants caution, although it may merit future investigation with longer datasets.



**Figure 7.** Geographical regions of influence of the major climate modes and corresponding cycles extracted from the wavelet GMPS. Left panels show the Pearson correlation (R) between rainfall and the relevant climate mode where it is over 0.1. Sites where the correlation is significant ($p < 0.05$) are circled in black. The right panels show the wavelet GMPS for each climate mode. The cycles found in global rainfall are coloured alike to the rainfall cycles in Section 4.1.1, other cycles are shown in grey.



**Figure 8.** Comparison of the fixed 12.9-, 19.9-, and 28.2-year GRCs with the extracted cycle from wavelet analysis of the major climate modes.(**a**) Strong phase alignment can be seen between the 12.9-year GRC, Niño 3.4 and IPO (R>0.9). (**b**) The 19.9-year GRC aligns well (R>0.8), but with the correlation weakening post-1960. (**c**) The ~28-year cycles in Niño 3.4 and IPO also fit closely to the 28.2 GRC although offset by 8-year.



## 5 Discussion and Conclusions

This study has identified three Global Rainfall Cycles (12.9-, 19.9-, 28.2-year) with coherent phase alignment in regions like Australasia and the Americas. Validation of our results at individual sites aligns well with existing studies (Mitra and Dutta, 1992; Williams et al., 2021). An overlap with the spatial distribution and interdecadal frequency of Niño 3.4 index and Interdecadal Pacific Oscillation (IPO) was identified. However, the variance explained by the 12.9- and 19.9-year GRCs at individual sites compared to the corresponding variance in ENSO and IPO suggest that these climate modes cannot fully account for the impact observed, pointing to the potential influence o**f an unknown external driver.**

### 5.1 Relative contributions of the GRCs and climate modes to rainfall

The observed GRCs show strong alignment with interdecadal cycles in the Niño 3.4 region of the Pacific Ocean and the IPO. Lower SST (i.e., La Niña conditions) and negative IPO phases are typically associated with increased rainfall in Australasia and reduced rainfall across specific regions in the Americas, particularly in latitudes between the tropics and the polar circles. This teleconnection is also evident in both the 12.9- and 19.9-year GRCs, suggesting that ENSO and the IPO are likely contributors to these rainfall patterns. However, the signal in rainfall at each site is far stronger than we would expect if ENSO were the sole driver of this phenomenon. The mean variance explained across significant sites of the 19.9-year GRC and the total influence of Niño3.4 on rainfall show similar values (10.99%, 10.62% respectively), even though the 19.9-year component of the climate mode only accounts for roughly 4% of variance. The same is true for the 12.9-year GRC and Niño 3.4 variance explained at significant sites (11.16%, 10.62% respectively), though the contribution of that cycle to the climate mode is only 11%. The coefficient of determination for the 12.9- and 19.9-year GRC in rainfall is equal to or exceeds that of the corresponding cycles within the Niño 3.4 and IPO climate modes. This indicates that their direct influence on rainfall is at least as strong as their contribution to these modes, which themselves explain only a portion of rainfall variance, and suggests they are not the sole drivers of these cycles.

Choosing a site with a strong ENSO signal allows us to visualise the discrepancy more clearly. Figure 9 compares the direct influence of the ENSO (Niño 3.4 index) to rainfall at a single site in north-eastern Australia. For this site the normalised 19.9-year GRC has a markedly smaller influence in the Niño 3.4 index than directly in the rainfall anomaly (Fig. 9d-f). If the sole mechanism of action was through the climate mode, we would expect the scale of rainfall anomaly to be approximately proportional to the cycle within ENSO, but we find the opposite. The 19.9-year GRC has a stable amplitude and periodicity as well as a strong SNR (0.81) in explaining the direct rainfall anomaly (Fig. 9.d), whereas the signal in Niño 3.4 shows a decreasing amplitude, with a mean SNR of 0.34 (Fig. 9e). The Pearson correlation of the 19.9-year cycle to Niño 3.4 (0.20) and rainfall at the site (0.24) are reasonably similar, and yet ENSO only accounts for 24% of the total rainfall variance ($R^2$). Hence, the 19.9-year GRC effect is stronger and cleaner in explaining rainfall than it is for Niño 3.4. We can expand this by looking at the spatial distributions of each.





**Figure 9.** Comparison of contributions of the relative strength of the 19.9-year cycle in rainfall at a selected site in Australia and the Niño 3.4 SST. (a) Normalised (z-score) rainfall anomaly for the site, complete time series with no infilling. (b) Site location on a global map. (c) The wavelet spectrum of annual rainfall.(d) The rainfall anomaly with the 19.9-year cycle extracted by wavelet analysis (orange), and the fixed 19.9-year GRC (green). (e) The rainfall anomaly shown with theNiño 3.4 SST time series (orange) scaled by linear regression to visualise contribution to rainfall, and the 19.9-year component of the Niño 3.4 SST time series (green). (f) the rainfall anomaly with the 19.9-year cycle extracted from rainfall (orange) and the theNiño 3.4 SST time series (green). Though the two cycles are mostly in phase, the GRC in rainfall has a much higher SNR (0.81 versus 0.34) as well a consistent amplitude and period.

The 12.9-year GRC shows clear overlap with the Niño 3.4 but notable divergences in their spatial distributions are also evident. The 12.9- and 28.2-year GRCs show clusters of significant spatial coherence in northern Canada and Greenland where the ENSO influence is weak or non-existent. The 19.9-year GRC also shows strong agreement with the IPO (Fig. 7b), which is easier to identify in the correlation to the extracted wavelet (Fig. 6e) due to the strength of the signal. South America, Africa, the western USA, northern Canada and Alaska all show consistent spatial and phase coherence. However, the 19.9-year GRC cycle also shows clustering around the American east coast, the Australian west coast, Brazil, and up





through eastern Asia. This reinforces the interpretation that a direct mechanism may be acting on rainfall as well as the climate modes where the influence would be amplified.

Climate modes are complex phenomena, defined by chaotic interactions of multiple drivers such as surface air 420 temperatures, wind patterns, solar insolation, subsurface temperature anomalies, the thermocline, humidity distribution, ocean currents and salinity. The interdecadal variability within them has long been thought to be quasi-periodic and essentially unpredictable (Power and Colman, 2006; Timmermann et al., 2018). This research suggests that the relatively weak but stable 12.9-, 19.9- and 28.2-year cycles within the El Niño Southern Oscillation are having a more significant and consistent impact on global rainfall than currently assumed.

## 425 5.2 Validation and alignment to previous research

The cycles presented here are consistent with previous regional studies on periodicity. If we compare the findings of Williams et al. (2021) in Sierra Nevada rainfall (significant cycle of 13- to 15-years; 21% of variance) to the nearest GPCC grid point (36.25°N, 118.75°W; Fig. 5), the results match closely. Here we see the 12.9-year GRC (Fig. 5e) with a similar variance of 14% and an SNR of 0.95. The cycle is 180° out of phase with Australasia, which is consistent with ENSO 430 teleconnection where increased rainfall in Australia coincides with lower rainfall on the west coast of the United States (Fig. 7a; Timmermann et al., 2018). The region also shows a cluster of significant sites on the global maps, negatively correlated to the 12.9-year GRC (Fig. 6a, d). This strongly supports the notion that the ~13-year cycle identified by Williams et al (2021) is the 12.9-year GRC identified here.

A similar consistency can be found for the 19.9-year GRC in regions were a ~20-year cycle was previously 435 attributed to the 18.6-year Lunar Nodal Cycle (LNC). Mitra and Dutta (1991) analysed 115 years of summer monsoonal rainfall (1871-1985) in the Assan macroregion of India. They attributed the cyclic signal to the LNC, but this required a phase inversion to stay aligned. In contrast, individual analysis of the closest site in the GPCC dataset (26.25°N, 91.25°E, Fig. S4 in the Supplement) shows the 19.9-year GRC consistently in phase to the extracted wavelet (R=0.85), accounting for 9% of rainfall variance and an SNR of 0.63. The region also shows up as significant and positively correlated in the global 440 maps (Fig. 6b).

## 5.3 Limitations of traditional significance testing in large-scale cycle detection

The GC-WAPS method allowed for the identification of subtle cyclic influence across a large number of sites. However, our results suggest that the common practice of using significance over red noise to detect individual cycles may be overly stringent for this type of analysis. The current method of identifying statistically significant frequencies was 445 developed largely in response to the criticism that wavelet analysis was susceptible to bias in subjective visual analysis of the spectra (Torrence and Compo, 1998), albeit with a focus on individual time series. However, when looking at large sets of data, there are other means to test whether specific cycles are occurring randomly - namely the number of sites showing cycles, their phase, and spatial distribution. These methods are of critical importance when dealing with climate systems





where the global data are frequently spread across multiple sites, noisy and incomplete. For example, if we filter the GPCC
v2022 dataset by the Torrence and Campo (1998) test for significance at 95%, exceeding the threshold for at least half the
time span (60 years), at each individual site for the 12.9-year cycle, we end up with only 15 significant sites of the full 3,639.
**These would be dismissed as Type I errors** and cause us to miss the other lines of evidence presented above. For an
individual time series, the significance test is still the most rigorous, but these results should also make us aware of its
limitations and the risk of Type II errors.

The GC-WAPS method provides an approximate overview of dominant cycles in large datasets but also requires
follow-up analysis due to inherent limitations. The automated peak selection from GMPS is sensitive to missing peaks that
may have a broad power, or a low signal to noise ratio. Regional comparison revealed the highest cycle density in
Australasia, with consistent evidence of the same cycles across all regions and no emergence of alternative periods. However
a more granular site by site analysis was still required to verify these cycles and the scale of their influence (Sect 4.2, 4.3).

## 5.4 Gridded versus gauged datasets

A validation of the method was also performed to assess whether transitioning from a purely gauged dataset to
gridded would substantially affect the results. Much work around periodicity in rainfall derived from the research of Robert
Currie, and he was adamant that any form of spatial averaging would mask the periodic signal (Currie and Vines, 1996). We
repeated the GC-WAPS analysis of the gridded GPCC dataset but limited to the region of eastern Australia studied by
Selkirk et al (2025) using the SILO gauged dataset. The results were nearly identical for the three significant cycles at ~13-,
~20- and ~30-year (Fig. S1 in the supplement). The GPCC data provided even clearer clustering of these cycles than did
Selkirk et al. (2025), with reduced spread (σ) and increased density of the Gaussian distributions while remaining highly
significant (p < 0.0001) over red noise. The improved clarity from evenly spaced gridded points, rather than a loss of signal,
was a positive outcome and suggested that the cycle detection is not significantly disrupted by spatial interpolation at this
scale using the GC-WAPS method.

## 5.5 Quantifying the influence

Accurately quantifying the global influence of interdecadal cycles is challenging. One the most common methods
for estimating the contribution of a cycle or driver to time series variance is the coefficient of determination ($R^2$) (Power et
al., 1999; Torrence and Compo, 1998). However, caution must be used in the interpretation for evaluating long-period
signals in high-variance fields like rainfall. The dominance of interannual noise can obscure the contribution of lower-
frequency oscillations, even when they have spatial coherence, a strong amplitude and climatic relevance. This can result in
underestimating the importance of meaningful signals, particularly in short or noisy records. Accordingly, reliance on $R^2$
alone risks Type II errors, and should be complemented by other metrics to gain a more complete picture.

Using the extracted wavelet for a given period can give a clear indication of the cyclic influence at a particular site,
yet this too can be misleading. Wavelet analysis combines truncated sine waves across the time series, capturing frequency





modulation inherent in natural cycles and potentially providing a clearer variance estimate. However, if these components consistently drift from the mean frequency toward longer or shorter periods, it may misrepresent the true cycle length. Thus, comparing the fixed sine wave with the extracted wavelet (Fig. 6d-f) offers a more reliable indicator of regions where the GRC periods remain stable over time. For the 12.9- and 19.9-year GRC the variance from extracted wavelet averaged over

all significant sites was nearly double that of the fixed wavelet (6.19:11.16%, and 5.78:10.99% respectively), indicating the marked difference between the two methods. We can assume the true variance may lie somewhere in-between these two values, though neither give an indication of the cycle amplitude. Even large amplitude cycles can yield low $R^2$ values when overshadowed by interannual noise. Metrics like signal-to-noise ratio (SNR) are therefore useful complements, capturing the magnitude of the signal relative to background variability.

The influence of the GRCs becomes far more apparent when assessed by amplitude rather than variance. Across all significant sites, the average SNR for the three cycles ranges from 0.83 to 0.91, which is nearly as large as the standard deviation of the underlying rainfall anomaly. SNR can be interpreted as a reflection of the strength of external forcing relative to internal variability(Feldstein, 2000). The observed disparity between the high SNR (~0.87) and low explained variance (~8%) mirrors the signal-to-noise paradox where real-world signals can exhibit coherent and climatically

meaningful patterns despite appearing statistically weak due to being embedded in noisy systems(Scaife and Smith, 2018)). This reinforces the findings of Sect. 5.1, indicating the probable presence of an unknown external driver, acting on both ENSO (through SST) and directly at the site.

## 5.6 Implications and future research

The implications of these findings are significant for the development of large-scale climate models and global

rainfall forecasting. Collaborative frameworks like the Coupled Model Intercomparison Project Phase 6 (CMIP6) underpin much of our understanding of climate behaviour, informing the work of scientists, policymakers, and organizations such as the Intergovernmental Panel on Climate Change (IPCC). If the 12.9- and 19.9-year cycles, in particular, are as spatially coherent and globally distributed as the analysis suggests, they offer a potential new dimension to improve the fidelity of these models. Even modest periodicities, when persistent, can shape multi-year drought and flood risks, modulate regional

climate modes, and influence the likelihood of rainfall extremes. Their inclusion could specifically strengthen seasonal-to-decadal rainfall forecasts in the regions of greatest GRCs influence.

The practical implications extend beyond academic research. Better characterisation of low-frequency rainfall cycles could inform agricultural planning, water resource management, and even insurance and reinsurance industries, where

understanding the timing and recurrence of rainfall extremes is critical for managing risk. Although the mechanism behind these periodicities remains unknown, their consistency suggests they may be the result of as-yet-unrecognized forcing in the climate system. Future research should adopt a broader systems-level perspective to explore their origins. Regardless of their

source, these cycles merit integration into both the scientific understanding and practical management of global rainfall variability.

**Code availability**

The PyWavelets package was used for decomposing the annual rainfall signal and is available though a Zenodo repository (https://zenodo.org/records/13306773; Lee et al., 2019). Visualisation of the wavelet spectrum was generated using *scaleogram* (https://github.com/PyWavelets/pywt; Sauvé and Nowacki, 2023).

**Data availability**

GPCC Full Data Monthly Product Version 2022 (2.5°) was used for global gridded rainfall analysis available from Deutscher Wetterdienst Website (https://opendata.dwd.de/climate_environment/GPCC/; Schneider et al., 2022). The gauged daily rainfall of eastern Australia was taken from the SILO point-based climate data set, available from the Queensland Government's SILO project  (https://www.longpaddock.qld.gov.au/silo/view-point-data/; Scientific Information for Land Owners, 2024). The National Oceanic and Atmospheric Administration Physical Science Laboratory landing page (https://psl.noaa.gov/data/timeseries/month/) was used to access Niño 3.4 Index (Rayner et al, 2024) and Indian Ocean Dipole data (Saji et al, 2024). The North Atlantic Oscillation timeseries was accessed through the NCEI (NOAA National Centers for Environmental Information, 2025). The Interdecadal Pacific Oscillation data was accessed through the Ministry for the Environment Data service (2017) of the New Zealand Government.

**Competing interests**

The authors declare that they have no conflict of interest.

**Financial Support**

This research was supported by an Australian Government Research Training Program (RTP) Scholarship.



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
