# Peer review of "Interdecadal rainfall cycles in spatially coherent global regions and their interaction with climate modes"

_EGUsphere, 2025_

## Author Response (AR1)

**RC1**

Thank you for taking the time to review this manuscript and provide valuable clarification. In the following response text from review comments is shown in bold and our response to each point are in plain text. Proposed changes to the manuscript will be given in quotation marks.

This study investigates the interdecadal rainfall cycles in spatially coherent global land regions using a Gaussian mixture model based on GPCC precipitation data, and further explores the relationships between the identified Global Rainfall Cycles (GRCs) and major climate modes including ENSO, IPO, IOD, and NAO. Compared with previous regional analysis, this innovative research provides a global distribution of GRCs and examines their links with large-scale climate modes, offering important implications for global water resource management and rainfall modelling. The study fits well within the scope of Hydrology and Earth System Sciences. After a careful review, I raise the following points for consideration:

1. How should the significant correlations between rainfall cycles in eastern Australia and NAO be interpreted (Fig.7d), given that the NAO is generally not considered a primary driver of rainfall variability in this region?

Yes, this correlation is intuitively unexpected. While NAO is not a primary driver in eastern Australia, studies have shown a teleconnection between eastern Australian rainfall and NAO over decadal timescales. The NAO can have an impact on sea surface temperatures (SST) in the Southern Ocean (SO). Long-term control simulations of an ocean-atmosphere coupled model suggested the NAO exerts a delayed effect on the Atlantic meridional overturning circulation generating an alternating effect in SST anomalies between the subpolar North Atlantic Ocean and the SO, eventually impacting subtropical eastern Australian rainfall (Sun et al., 2015).

This is one of several correlations seen in the paper where rainfall appears related to a climate mode outside its supposed zone of influence. Another is IPO clusters in the upper northern latitudes (Fig. 7b.), confirmed by a recent study which proposed tropical and extratropical Pacific decadal variability can explain up to half of observed decadal surface temperature trends in the Arctic (Svendsen et al., 2021). This indicates the inherently interconnected nature of the climate modes but also the possibility of higher level drivers which may be influencing the interdecadal component of each as well as site specific rainfall.

**Source:**

Sun, C., Li, J., Feng, J., and Xie, F.: A Decadal-Scale Teleconnection between the North Atlantic Oscillation and Subtropical Eastern Australian Rainfall, Journal of Climate, 28, 1074–1092, https://doi.org/10.1175/JCLI-D-14-00372.1, 2015.

Svendsen, L., Keenlyside, N., Muilwijk, M., Bethke, I., Omrani, N.-E., and Gao, Y.: Pacific contribution to decadal surface temperature trends in the Arctic during the twentieth century, Clim Dyn, 57, 3223–3243, https://doi.org/10.1007/s00382-021-05868-9, 2021.

The manuscript was updated to reflect this in Sect. 4.4 of the Results (line 360, page 17):

"Although there are cycles in the NAO and IOD, these bear little relationship to the GRCs. The IOD shows an 11.5-year and the NAO a 14.1-year peak (Fig. 7f) but neither of these align well to the 12.9-year GRC in phase or period when directly compared (Fig. S3 in the Supplement.). The correlation between eastern Australian rainfall and the NAO seems unexpected (Fig. 7d), while not a primary driver studies have shown a teleconnection between eastern Australian rainfall and NAO over decadal timescales (Sun et al., 2015). The cluster of IOD influence along the Great Australia Bight (south coast) is not reflected in any of the GRC distributions.

There is a consistent ~45-year cycle across the IPO, IOD and NAO climate modes (Fig.7f-h), similar to the 45.4-year cycle identified by GC-WAPS (Fig. 2). The length of this cycle relative to the time series (~130 years) and its position near the practical COI limit (~50 years; Sect. 3.1) warrants caution, although it may merit future investigation with longer datasets as it suggests the possibility of higher level drivers which may be influencing the interdecadal component of each."

Note: Figure 7 was also modified to remove the colour from the GMPS for Nino 3.4 and IPO (Fig.7e-f). The use of blue and red to delineate the cycles as opposed to the use for positive and negative correlation in the adjacent map was thought to be needlessly confusing.

2. The results show that the temporal consistencies in amplitude between the fixed 12.9- and 19.9- year GRCs with the extracted cycles from wavelet analysis of major climate modes (ENSO, IPO) deteriorate after the 1980s (Fig.8 a and b). Could you discuss possible causes for this? What implications can be drawn from this change?

This is a keen observation raised by both reviewers and there may be a few reasons for this:

- 1. From a purely analytical perspective we would expect a diminishing amplitude in the extracted wavelet as a result of the Cone of Influence for wavelet analysis. The timeseries is padded with zeros as the wavelet extends past the boundary. This is proportional to the period by a scale of  $\sqrt{2}$  (Torrence & Compo, 1998). This can be seen in the 12.9-year and 28.2-year wavelets taken from the Niño3.4 and the IPO (Fig. 8a and c), with a fall in amplitude as they approach the start and end boundaries. However this does not appear to account for the effect observed in the 19.9-year cycle which shows a more steady decline.
- 2. Recent studies demonstrated a marked increase in ENSO variability (standard deviation and amplitude) over the 1950s-1990s, followed by a sharp fall post 2000 (Fedorov et al., 2020). The authors suggested that changes such as the warming of the Indian Ocean and Atlantic multidecadal variability can strengthen the Walker circulation and dampen ENSO along with internal modulation producing multi-decadal swings in amplitude. Given that the 19.9-year signal in Niño3.4 was quite weak (accounting for roughly 4% of variance) it is not surprising to see the extracted wavelet amplitude impacted by these changes. All data has been detrended before analysis but this may be an indication of the possible effects of the changing global climate on these cycles.
- 3. Previous authors who have observed interdecadal cycles rainfall have frequently noted a varying degrees of amplitude and frequency modulation over extended periods (Currie, 1995; Williams et al., 2021). It is difficult to speculate on the cause of this without a better idea of what the drivers may be. It could be a variation in the strength of the interdecadal signal, or an increase in the variability of the more chaotic interannual component as noted above. It is a limitation of only having 130 years of data, making it challenging to identify trend versus modulation over longer periods.

The diminishing amplitude and varying period in the 19.9-year cycle is further evidence that the effect observed in Australian rainfall does not occur solely through ENSO/IPO, as the cycle is shown to have a continuous impact on rainfall in Figure 9d over the last 130 years. However, the paper which focussed on these cycles specifically in eastern Australia (Selkirk, 2025) also noted a differing modulation in effect across some sites. This would become an important factor if considering the use of the cycles for long-term forecasting.

**Sources:**

Currie, R. G.: Luni-solar and solar cycle signals in lake Saki varves and further experiments, Int. J. Climatol., 15, 893–917, https://doi.org/10.1002/joc.3370150805, 1995.

Fedorov, A. V., Hu, S., Wittenberg, A. T., Levine, A. F. Z., and Deser, C.: ENSO Low-Frequency Modulation and Mean State Interactions, in: Geophysical Monograph Series, edited by: McPhaden, M. J., Santoso, A., and Cai, W., Wiley, 173–198, https://doi.org/10.1002/9781119548164.ch8, 2020.

Williams, A. P., Anchukaitis, K. J., Woodhouse, C. A., Meko, D. M., Cook, B. I., Bolles, K., and Cook, E. R.: Tree Rings and Observations Suggest No Stable Cycles in Sierra Nevada Cool□Season Precipitation, Water Resources Research, 57, e2020WR028599, https://doi.org/10.1029/2020WR028599, 2021.

The manuscript was updated to reflect this in Sect. 4.4 of the Results (line 339, page 17):

"This weak signal may be related to the diminishing correlation of the 19.9-year cycle after the 1960's (Fig.8b) addressed in the discussion. The effect of the Cone of Influence on wavelet amplitude can be observed on the extracted 12.9- and 28.2-year cycles (Fig.8a, c) with a roll off in amplitude as they approach the start and end boundaries."

And further in Sect. 5.2 of the Discussion (line 419, page 21):

"The 19.9-year GRC has a stable amplitude and periodicity as well as a strong SNR (0.81) in explaining the direct rainfall anomaly (Fig. 9.d), whereas the signal in Niño 3.4 shows a decreasing amplitude, with a mean SNR of 0.34 (Fig. 8b; Fig. 9e). Recent studies demonstrated a marked increase in ENSO variability (standard deviation and amplitude) over the 1950s-1990s, followed by a sharp fall post 2000 (Fedorov et al., 2020). The authors suggested that changes such as the warming of the Indian Ocean and Atlantic multi-decadal variability can strengthen the Walker circulation and dampen ENSO along with internal modulation producing multi-decadal swings in amplitude. The weakening signal in ESNO may be an indication of the possible effects of the changing global climate on these cycles, even though it is not observed directly in rainfall. The Pearson correlation of the 19.9-year cycle to Niño 3.4 (0.20) and rainfall at the site (0.24) are reasonably similar, and yet ENSO only accounts for 24% of the total rainfall variance (R2). Hence, the 19.9-year GRC effect is stronger and cleaner in explaining rainfall than it is for Niño 3.4."

**3. This study suggests the potential influence of unknown external drivers on rainfall cycles beyond the well-known major climate modes. In your view, what types of external drivers might be included? Could land-surface conditions (e.g. soil, land use changes) also play a role?**

When this paper was submitted we did not have a viable driver that aligned to the cycles observed. We have since identified a potential mechanism. We are writing this up as a separate manuscript as it requires substantial evidence to support the hypothesis and would be too long to include in this paper (which is already ~8,000 words). Regional factors such as land use changes may indeed play a role on site specific impacts; however it is hard to see how they would lead to periodicity in rainfall across so many different regions.

The manuscript was updated to reflect this rationale with the following text in the Section 5.6 Implications for future research (line 524, page 25):

"However, before they can be incorporated into such forecasting a viable driver and mechanism would need to be proposed. The next step would be to try and observe these cycles in connected climate variables which may lend weight to their effect and provide a pathway to their source."

**4. Is "interaction" the most appropriate term to describe the relationship between GRCs and climate modes? Would "relationship" or "linkage" be more accurate in this context?**

A good point, I think "relationship" may be a better choice in context of the full paper and will be changed in the manuscript.

Revised manuscript title:

"Interdecadal rainfall cycles in spatially coherent global regions and their relationship to the climate modes"

**5. Please check and verify that all figure citations are correct in the second paragraph of Section 4.4.**

Thank you for pointing this out, the manuscript was updated following text in the Sect. 4.4 (line 333, page 17):

"The IPO shows the strongest spectral alignment with the GRCs, as GMPS peaks at 12.8- and 19.5-year (Fig. 7f), closely match the 12.9- and 19.9-year GRCs. Its influence spans Australasia, North America, and South America (Fig. 7b), mirroring the GRC's distribution (Fig. 6b, e). Time series analysis confirms this, with the 12.9- and 19.9-year GRCs showing high correlations (R = 0.93) with IPO-derived wavelet cycles, although 180° out of phase (Fig. 8a, b). This is to be expected as the IPO index is focussed on the interdecadal component of SST. The 19.9-year component of the Niño 3.4 index can be seen in the GMPS (Fig. 7e) but the extracted wavelet (Fig 8.b) shows how weak the signal is, contributing only 4% of temperature variance."

6. The structure of discussion could be improved. It is suggested that Section 5.1 and 5.2 be interchanged, as suggested by the logic in the first paragraph of Section 5. Section 5.3, 5.4, and 5.5 all address the strengths and limitations of the methods used in this study, and might be more effective if combined into a single section.

This is an excellent point, Section 5.2 on validation does make more sense coming first and has been adjusted in the manuscript. Sections 5.3-5.5 could be combined to logically combined under a single header however we believe retaining the headings helps the reader navigate to the relevant part. Happy to combine at the editors recommendation.

**RC2**

The study identified three widespread global rainfall cycles (12.9-, 19.9-, and 28.2-year) by analyzing global precipitation data and found that the two shorter cycles are closely related to the ENSO and IPO climate modes. The results revealed that the influence of these cycles on rainfall variability exceeds what can be explained by the effects of these known climate modes alone, suggesting the potential existence of an unknown common driving mechanism. This discovery holds promises for improving global water management and flood-drought forecasting capabilities, while some issues should be addressed before accepting:

1. The subtitle in Line 210, "4.1.1GMM Clusters of the Global Rainfall Cycles (GRCs)" should be revised as 4.1 GMM Clusters of the Global Rainfall Cycles (GRCs)

Thank you for picking up on this, the spacing has been adjusted.

2. Please clarify if homogeneity testing was performed to address potential systematic biases in the GPCC data arising from evolving measurement techniques and station relocations over the 130-year record.

This is a good point on data quality control. We did not run any independent homogeneity testing. This is because the GPCC Full Data Monthly product (v2020) already applies multi-stage quality control and harmonisation (Becker et al., 2013). They apply parallel storage of multiple data sources to enable cross-checks, metadata reconciliation, and quality assessment prior to interpolation to the analysis grid (Schneider et al., 2022).

Our study targets interdecadal variability in annual rainfall after gridding. Any remaining station-level discontinuities should tend to dampen the cyclic signal rather than create coherent multi-decadal cycles across continents. We also took the following steps to reduce sensitivity to inhomogeneities:

- analysed gridded anomalies
- filtered by gauge coverage
- validated eastern-Australia results directly against the SILO gauge dataset, which showed the two data sets were in strong agreement (Section 5.4, line 460)

Together, these choices make additional station-by-station homogenization out of scope for this paper and unlikely to alter our conclusions.

**Sources:**

Becker, A., Finger, P., Meyer-Christoffer, A., Rudolf, B., Schamm, K., Schneider, U., and Ziese, M.: A description of the global land-surface precipitation data products of the Global Precipitation Climatology Centre with sample applications including centennial (trend) analysis from 1901–present, Earth System Science Data, 5, 71–99, https://doi.org/10.5194/essd-5-71-2013, 2013.

Schneider, U., Hänsel, S., Finger, P., Rustemeier, E., and Ziese, M.: GPCC Full Data Monthly Version 2022 at 2.5°: Monthly Land-Surface Precipitation from Rain-Gauges built on GTS-based and Historic Data: Globally Gridded Monthly Totals, [data set] (2022), https://doi.org/10.5676/DWD GPCC/FD M V2022 250, 2022.

The manuscript was updated to address this with the following text in Sect. 2 (line 86, page 8):

"No independent homogeneity testing was performed in addition to the multi-stage quality control and harmonisation applied in the dataset preparation (Becker et al., 2013)."

3. Please note that the 2.5° gridded data product is based on the spatial interpolation of gauge observations, which may introduce uncertainties, particularly in data-sparse regions such as oceans, high latitudes, and parts of the tropics. It's recommended to clarify this inherent limitation of the dataset and briefly discuss its potential implications for the interpretation of the identified global rainfall cycles.

Yes, another good point. GPCC's 2.5° fields are produced by interpolating land-only station anomalies using a modified SPHEREMAP scheme on a high-resolution sub-grid and then aggregating to coarser grids. GPCC identifies three dominant uncertainty sources for the gridded analyses:

- i. Systematic gauge bias (e.g., wind under-catch)
- ii. Stochastic sampling error from sparse/uneven station density
- iii. Residual errors (spatial and temporal discontinuities of precipitation measurements)

Uncertainty fields and corrections are provided by gridded point but were not used in this analysis. This was mainly because they were difficult to incorporate into the analysis method so we sought to mitigate them in three ways:

- i. GC-WAPS analysis was limited to ≥90% gauge-informed months
- ii. We worked with annual anomalies, reducing high frequency sampling noise and gauge biases
- iii. By showing the strongest signals occurred in data rich regions and can be replicated by an independent point-gauge dataset over eastern Australia (SILO)

The manuscript was updated to address this rationale with the following text in Sect. 2 (line 98, page 5):

"GPCC identifies three dominant uncertainty sources for the gridded analyses: systematic gauge bias, stochastic sampling error from sparse/uneven station density and residual errors (spatial and temporal discontinuities of precipitation measurements). Uncertainty fields and corrections are available but were not used in this analysis as we sought to mitigate their influence by limiting the inputs by filtering gauge-informed months, reducing sampling noise by aggregation to annual data, and comparison to independent point-gauge datasets over eastern Australia (see Sect 3.1)."

**And further in Sect. 5.4 of the Discussion:**

"A validation of the method was also performed to assess whether transitioning from a purely gauged dataset to gridded would substantially affect the results. Much work around periodicity in rainfall derived from the research of Robert Currie, and he was adamant that any form of spatial averaging would mask the periodic signal (Currie and Vines, 1996). This also helped to address the uncertainties arising from the spatial interpolation of the gauge observations in GPCC v2020. We repeated the GC-WAPS analysis of the gridded GPCC dataset but limited to the region of eastern Australia studied by Selkirk et al (2025) using the SILO gauged dataset. The results were nearly identical for the three significant cycles at  $\sim$ 13-,  $\sim$ 20- and  $\sim$ 30-year (Fig. S1 in the supplement). The GPCC data provided even clearer clustering of these cycles than did Selkirk et al. (2025), with reduced spread ( $\sigma$ ) and increased density of the Gaussian distributions while remaining highly significant (p < 0.0001) over red noise. The improved clarity from evenly spaced gridded points, rather than a loss of signal, was a positive outcome and suggested that the cycle detection is not significantly disrupted by spatial interpolation, or its inherent uncertainty, at this scale using the GC-WAPS method."

4. It is noted that the comparison between the fixed 12.9- and 19.9-year GRCs and the corresponding cycles extracted from wavelet analysis of major climate modes shows some discrepancies after 1980, particularly in amplitude and phase. Could you please explain the possible reasons for these inconsistencies? Could they be influenced by other factors?

This is a keen observation raised by both reviewers and there may be a few reasons for this:

- 1. From a purely analytical perspective we would expect a diminishing amplitude in the extracted wavelet as a result of the Cone of Influence for wavelet analysis. The timeseries is padded with zeros as the wavelet extends past the boundary. This is proportional to the period by a scale of  $\sqrt{2}$  (Torrence & Compo, 1998). This can be seen in the 12.9-year and 28.2-year wavelets taken from the Niño3.4 and the IPO (Fig. 8a and c), with a fall in amplitude as they approach the start and end boundaries. However this does not appear to account for the effect observed in the 19.9-year cycle which shows a more steady decline.
- 2. Recent studies demonstrated a marked increase in ENSO variability (standard deviation and amplitude) over the 1950s-1990s, followed by a sharp fall post 2000 (Fedorov et al., 2020). The authors suggested that changes such as the warming of the Indian Ocean and Atlantic multidecadal variability can strengthen the Walker circulation and dampen ENSO along with internal modulation producing multi-decadal swings in amplitude. Given that the 19.9-year signal in Niño3.4 was quite weak (accounting for roughly 4% of variance) it is not surprising to see the extracted wavelet amplitude impacted by these changes. All data has been detrended before analysis but this may be an indication of the possible effects of the changing global climate on these cycles.
- 3. Previous authors who have observed interdecadal cycles rainfall have frequently noted a varying degrees of amplitude and frequency modulation over extended periods (Currie, 1995; Williams et al., 2021). It is difficult to speculate on the cause of this without a better idea of what the drivers may be. It could be a variation in the strength of the interdecadal signal, or an increase in the variability of the more chaotic interannual component as noted above. It is a limitation of only having 130 years of data, making it challenging to identify trend versus modulation over longer periods.

The diminishing amplitude and varying period in the 19.9-year cycle is further evidence that the effect observed in Australian rainfall does not occur solely through ENSO/IPO, as the cycle is shown to have a continuous impact on rainfall in Figure 9d over the last 130 years. However, the paper which focussed on these cycles specifically in eastern Australia (Selkirk, 2025) also noted a differing modulation in effect across some sites. This would become an important factor if considering the use of the cycles for long-term forecasting.

**Sources:**

Currie, R. G.: Luni-solar and solar cycle signals in lake Saki varves and further experiments, Int. J. Climatol., 15, 893–917, https://doi.org/10.1002/joc.3370150805, 1995.

Fedorov, A. V., Hu, S., Wittenberg, A. T., Levine, A. F. Z., and Deser, C.: ENSO Low-Frequency Modulation and Mean State Interactions, in: Geophysical Monograph Series, edited by: McPhaden, M. J., Santoso, A., and Cai, W., Wiley, 173–198, https://doi.org/10.1002/9781119548164.ch8, 2020.

Williams, A. P., Anchukaitis, K. J., Woodhouse, C. A., Meko, D. M., Cook, B. I., Bolles, K., and Cook, E. R.: Tree Rings and Observations Suggest No Stable Cycles in Sierra Nevada Cool-Season Precipitation, Water Resources Research, 57, e2020WR028599, https://doi.org/10.1029/2020WR028599, 2021.

The manuscript was updated to reflect this in Sect. 4.4 of the results (line 339, page 17):

"This weak signal may be related to the diminishing correlation of the 19.9-year cycle after the 1960's (Fig.8b) addressed in the discussion. The effect of the Cone of Influence on wavelet amplitude can be observed on the extracted 12.9- and 28.2-year cycles (Fig.8a, c) with a roll off in amplitude as they approach the start and end boundaries."

And further in Sect. 5.2 of the discussion (line 419, page 21):

"The 19.9-year GRC has a stable amplitude and periodicity as well as a strong SNR (0.81) in explaining the direct rainfall anomaly (Fig. 9.d), whereas the signal in Niño 3.4 shows a decreasing amplitude, with a mean SNR of 0.34 (Fig. 8b; Fig. 9e). Recent studies demonstrated a marked increase in ENSO variability (standard deviation and amplitude) over the 1950s-1990s, followed by a sharp fall post 2000 (Fedorov et al., 2020). The authors suggested that

changes such as the warming of the Indian Ocean and Atlantic multi-decadal variability can strengthen the Walker circulation and dampen ENSO along with internal modulation producing multi-decadal swings in amplitude. The weakening signal in ESNO may be an indication of the possible effects of the changing global climate on these cycles, even though it is not observed directly in rainfall."